# Physical Health in Clinical High Risk for Psychosis Individuals: A Cross-Sectional Study

**DOI:** 10.3390/brainsci13010128

**Published:** 2023-01-12

**Authors:** Umberto Provenzani, Andrea De Micheli, Stefano Damiani, Dominic Oliver, Natascia Brondino, Paolo Fusar-Poli

**Affiliations:** 1Department of Brain and Behavioural Sciences, University of Pavia, 27100 Pavia, Italy; 2Early Psychosis, Interventions and Clinical-Detection (EPIC) Lab, Department of Psychosis Studies, Institute of Psychiatry, Psychology & Neuroscience, King’s College London, London SE5 8AF, UK; 3OASIS Service, South London and Maudsley NHS Foundation Trust, London SE11 5DL, UK; 4Maudsley Biomedical Research Centre, National Institute for Health Research, South London and Maudsley NHS Foundation Trust, London SE5 8AF, UK

**Keywords:** physical health, psychosis, risk, CHR-P

## Abstract

Background: The clinical high risk for psychosis (CHR-P) phase represents an opportunity for prevention and early intervention in young adults, which also could focus on improving physical health trajectories. Methods: We conducted a RECORD-compliant clinical register-based cohort study. The primary outcome was to describe the physical health of assessed CHR-P individuals, obtained via Electronic Health Records at the South London and Maudsley (SLaM) NHS Foundation Trust, UK (January 2013–October 2020). Results: The final database included 194 CHR-P subjects (46% female). Mean age was 23.70 ± 5.12 years. Percentage of tobacco smokers was 41% (significantly higher than in the age-matched general population [24%]). We found that 49% of subjects who consumed alcohol had an AUDIT-C (Alcohol Use Disorder Identification Test) score above 5 (hazardous drinking), with an average score of 4.94 (significantly higher than in the general population [2.75]). Investigating diet revealed low fiber intake in most subjects and high saturated fat intake in 10% of the individuals. We found that 47% of CHR-P subjects met the UK recommended physical activity guidelines (significantly lower than in the general population [66%]). Physical parameters (e.g., weight, heart rate, blood pressure) were not significantly different from the general population. Conclusions: This evidence corroborates the need for monitoring physical health parameters in CHR-P subjects, to implement tailored interventions that target daily habits.

## 1. Introduction

The clinical high risk for psychosis (CHR-P) construct enables the identification of individuals who have a greatly increased risk of developing a first episode of psychosis within 1–2 years compared to the general population [1,2]. The prevalence of CHR-P individuals in the community is still undefined [3] and there is scarce knowledge about the outcomes of individuals who do not transition to psychosis [4], who might remain at a lower level of functioning compared to non-psychiatric subjects [5]. In those individuals who will transition to psychosis, most will develop a schizophrenia spectrum disorder according to the DSM/ICD [6]. A 15-year follow-up study found that CHR-P individuals develop a psychotic disorder up to 10 years after initial presentation [7], in line with a risk of transition ranging from 65 to 79% at 10 years, reported by other studies [8]. In addition to attenuated psychotic symptoms (APS), brief limited intermittent psychotic symptoms (BLIPS) and genetic risk and deterioration syndrome (GRD) that define the construct [9], many CHR-P individuals often present other psychiatric comorbidities (i.e., anxiety, depression, substance abuse) that might be clinically debilitating [10,11]. Moreover, suicidality might be increased [12].

Another core feature of the CHR-P state is the marked impairment in psychosocial functioning, associated with high-risk symptoms [13]. Social impairment in these individuals tends to be resistant to pharmacological and psychosocial treatment [14], constituting a predictor of transition to psychosis [15,16] and is also associated with a decreased subjective quality of life [17,18]. Further deficits have also been underlined in the domain of social cognition, within a context of widespread mild cognitive deficits, falling at an intermediate level between that of healthy individuals and those diagnosed with schizophrenia [13].

The CHR-P phase thus represents an opportunity for prevention and early intervention in young adults (aged 18–35 years), which might also be focused on important physical ill-health trajectories in this group [19]. In fact, CHR-P individuals display a higher prevalence of cardiometabolic risk factors, compared with age-matched controls from the general population (e.g., increased blood pressure, waist circumference and fasting blood glucose) [20]. This vulnerability has been associated with modifiable physical health behaviors in CHR-P, such as reduced physical activity and increased rates of smoking and alcohol abuse [21]. There are different reasons to promote good physical health and lifestyles in CHR-P individuals. Recent evidence shows that low levels of physical activity during childhood and adolescence could be an independent predictor of psychosis in adulthood [22]. Hence, physical health interventions might directly reduce the risk of developing a psychotic disorder. Second, a large proportion of CHR-P individuals will develop psychiatric disorders other than psychosis (e.g., mood, anxiety and substance use disorders) [23], which are also linked to a significant reduction in daily functioning and physical health [24,25] related to these disorders. Finally, in those that will develop psychosis, intervening at the earliest stage with the adoption of a preventative approach is associated with better long-term outcomes [26], since psychosis is commonly associated with a wide range of comorbid and multiple physical-health conditions [27] and often progresses to chronic, severe conditions [28].

Despite these considerations, which underline that promoting physical health in these individuals might be beneficial, physical health outcomes are often not monitored even in specialized services for CHR-P [29], which is a problem shared with generic psychiatric services [30]. High-quality research focused on physical health monitoring and interventions in young people with CHR-P is still scarce [29], but very recent evidence shows that exercise in these individuals might improve not only fitness, but also cognitive performance and severity of attenuated positive psychotic symptoms [31]. Also, the development of acceptable and effective interventions in CHR-P is linked to the necessity of understanding more clearly why these individuals have poorer lifestyle profiles compared to individuals who are not at risk for psychosis [32].

### Aims of the Study

The primary aim of our study is to describe physical health and lifestyle outcomes of CHR-P individuals by routinely collecting physical health measures that are often neglected in this population (e.g., diet, physical activity). Secondary aims are the comparison of data with evidence collected from the general population (UK national averages) and investigation of gender differences. We hypothesize that CHR-P individuals will have poor physical health and lifestyle outcomes (e.g., higher tobacco and alcohol consumption, worse diet, less physical activity) and that these outcomes will be lower if compared to the general population. This evidence might deepen our knowledge of these important outcomes in the CHR-P phase, so that it can be used in the future for prevention and interventions focused on physical health in this group.

## 2. Materials and Methods

### 2.1. Participants and Study Design

This study is a clinical cross-sectional study using Electronic Health Records (EHRs), including EHR data on routine physical health checks [33] from all individuals from January 2013 to October 2020, managed by the South London and Maudsley (SLaM) National Health Service Foundation Trust, UK. The data source EHR employed in the current study provides contemporaneous EHR and ‘real-world’ data on routine mental healthcare from all patients managed by the South London and Maudsley (SLaM) NHS Foundation Trust. SLaM is a UK National Health Service (NHS) mental health trust that provides secondary mental healthcare to a population of 1.36 million individuals in South London (Lambeth, Southwark, Lewisham and Croydon boroughs). In SLaM, there is one of the highest rates of psychosis in the world [34]. In terms of the quality of SLaM/CRIS records, SLaM was an early pioneer of EHR and the trust is effectively digitized and paper-free. SLaM has a near-monopoly in terms of secondary mental healthcare provision to its local catchment area, and it is a legal requirement for SLaM healthcare professionals to keep these records up to date [35]. Whereas many national registers capture only those patients who have been hospitalized, the SLaM EHR register contains the full clinical records of all patients, which are continually updated throughout their care, regardless of discharges from and/or referrals to other services.

Study participants were assessed by OASIS (Outreach and Support in South London), which was set up in 2001 and it is one of the oldest early detection CHR-P services in the UK [36,37]. The service is focused on the identification, prognostic assessment, treatment (pharmacological, psychological, psychoeducational), and clinical follow-up of help-seeking CHR-P individuals aged 14–35 years, serving the SLaM catchment area. OASIS is integrated into the Pan-London Network for Psychosis-prevention (PNP) [38]. The study population included a sample of all individuals accessing OASIS in the period from January 2013 to October 2020, assessed with Comprehensive Assessment of At Risk Mental State (CAARMS) [9] and meeting CHR-P criteria: Brief Limited Intermittent Psychotic Symptoms (BLIPS), Attenuated Psychosis Symptoms (APS), and Genetic Risk and Deterioration Syndrome (GRD). All OASIS staff undergo extensive psychometric training to ensure high reliability in the designation of at-risk cases [39]. The OASIS population can be considered representative of the general CHR-P sample, since the level of risk enrichment observed (pretest risk [40]: 14.6% at more than 3 years [41]) aligns with that observed in CHR-P services worldwide (meta-analytical pretest risk 15% at more than 3 years [42]). 

### 2.2. Assessment Instruments

Baseline assessment of CHR-P subjects includes a routinary and comprehensive medical examination for physical parameters, which is complemented by validated questionnaires (see “Variables”), in line with the NICE (National Institute for Health and Care Excellence) Clinical Guideline 178 [43].

Fagerström Test for Nicotine Dependence (FTND) [44] is a standardized instrument consisting of 6 questions exploring daily cigarette consumption, compulsive use, and dependence. The score ranges from 0 to 10 (with higher scores indicating a most severe level of dependence to nicotine). More precisely, scorings from 0 to 2 indicate a low level of dependence, from 3 to 4 low-moderate dependence, from 5 to 7 moderate dependence, and more than 8 a high level of dependence. For people that use other types of nicotine consumption other than cigarette smoking (e.g., e-cigarette, nicotine gum, or nicotine patches), we have investigated habits and reported information in adapted versions of FTND already used in the literature (i.e., equivalence of 10 vape nicotine puffs for a cigarette [45] or a re-worded test for gum users [46]).AUDIT (Alcohol Use Disorder Identification Test) [47] consists of 10 self-administered questions to investigate alcohol use disorder. When AUDIT-C score, which includes core questions regarding alcohol units consumed and frequency of drinking, is equal or above 4, it might indicate hazardous drinking. Regarding the AUDIT total score, a low level of risk is identified with an overall score between 0 and 7, the range from 8 to 15 is the most appropriate for simple advice focused on the reduction of drinking. Higher scores (up to 19) suggest the need for brief counselling and continuous monitoring, while a complete diagnostic evaluation for alcoholic dependence is warranted for scores 20 and over.DINE (Dietary Instrument for Nutritional Education) [48] is a structured interview investigating the intake of dietary fiber and fat (unsaturated and saturated). Scores for fibers and fat are rated into 3 different categories: low (under 30), medium (between 30 and 40), and high intake (more than 40). Scores for unsaturated fat are rated as low (less than 6), medium (6 to 9), and high (more than 9)IPAQ (International Physical Health Questionnaire) [49] rates the level of physical activity. This tool comprises 3 different categories of physical activity based on the intensity (vigorous, moderate, and walking) and quantifies the amount of time spent sitting.

### 2.3. Variables

Baseline descriptive variables included:(1)Sociodemographic parameters: age, sex, ethnicity(2)Physical health data:
2.1Tobacco use: tobacco smoker status (yes/no), number of daily cigarettes, FTND score;2.2Alcohol use: alcohol drinker status (yes/no), AUDIT-C and AUDIT total score;2.3Type of diet: DINE total score, DINE fiber score, DINE fat score, DINE unsaturated fat score;2.4Physical Activity: IPAQ total score (MET), IPAQ vigorous, moderate and walking activity (days per week; minutes per day), IPAQ sitting (minutes per week)2.5Physical parameters: weight in kilograms, height in meters, Body Mass Index (BMI), waist circumference in centimeters, heart rate in beats per minute (bpm), respiratory rate in acts per minute (apm), systolic and diastolic pressure in mmHg.

### 2.4. Statistical Analysis

This clinical register-based cohort study was conducted according to the REporting of studies Conducted using Observational Routinely collected health Data (RECORD) Statement [50] (Appendix A). The primary outcome was to describe physical health data (see “variables”) in the sample. Sociodemographic parameters and physical health data (including missing data) were described with mean and SD for continuous variables, and absolute and relative frequencies for categorical variables. We employed the use of Student’s *t*-test for independent samples of numerical variables (i.e., number of cigarettes smoked, FTND score in smokers, AUDIT-C and AUDIT total score in drinkers, DINE total score, IPAQ score, weight, height, Body Mass Index, waist circumference, heart rate, respiratory rate, systolic pressure, diastolic pressure). Fisher’s exact test was employed for categorical variables (i.e., number of tobacco smokers and alcohol drinkers). A secondary aim was to compare the physical health data with the national average in the general population: we compared data from our sample with values taken from UK Office for National Statistics census data, referring to the same age-span (18–35) and the same period (2013–2019). The 95% confidence intervals were computed for our sample data using bootstrapping (10,000 samples). Differences were considered to be statistically significant if the census value did not fall within the bootstrapped 95% confidence intervals. Another secondary aim was to detect differences between male and female CHR-P subjects. For all analyses, statistical tests were two-sided and statistical significance was defined as *p* < 0.05, except for multiple *t*-test comparisons where we adjusted statistical significance using Bonferroni correction (*p* values are reported unadjusted). All analyses were conducted in IBM SPSS 28.0.

## 3. Results

### 3.1. Sample Characteristics

The final database included 194 CHR-P subjects, 90 (46%) females and 104 (54%) males. Mean age was 23.70 ± 5.12 years. The majority of the sample comprised white (41%) and black British (21%) subjects. Sociodemographic parameters are described in Table 1.

### 3.2. Physical Health Data in CHR-P Samples

#### 3.2.1. Tobacco Use

Tobacco smokers in the sample were 80 (41%, bootstrapped 95%CI 35–48%). Among the subjects who smoked, the mean number of cigarettes smoked every day was 9 ± 8 (bootstrapped 95%CI 7–11), the mean FTND score was 2.51 ± 2.54 (bootstrapped 95%CI 1.95–3.06). Overall, 64 subjects (33%) had a low to moderate level of dependence (FTND score ≤ 5), 12 subjects (6%) had a moderate level of dependence (FTND score between 5 and 7), and 4 subjects (2%) had a high level dependence (FTND score ≥ 7) (Figure 1). 

UK government statistics published in 2018 [51] concerning smoking habits in the general population reported that 24% of the subjects were smokers in the same age group (16–24), with an average consumption of 11 cigarettes a day. Average FTND score in UK, for the general population, has been reported as 3.00 [52]. Therefore, our sample reported a percentage of smokers that was significantly higher and almost double compared to the general population, while the number of cigarettes smoked daily and average FTND score in smokers were comparable.

#### 3.2.2. Alcohol Use

Alcohol drinkers in the sample were 139 (72%, bootstrapped 95%CI 66–77%). Among drinkers, the average AUDIT-C score was 4.94 ± 2.93 (bootstrapped 95%CI 4.43–5.45) and the average AUDIT total score was 7.88 ± 6.63 (bootstrapped 95%CI 6.80–8.99). Of note, 49% of the drinkers in our sample had an AUDIT-C score above 5 (cut-off for hazardous drinking) and 32% had an AUDIT total score above 8 (cut-off for advice to reduce alcohol) (Figure 1).

UK government statistics published in 2017 [53] reported a higher proportion of drinkers (80%) in the same age-group (16–24) in the general population. However, proportions of hazardous drinking were lower: 1.5% were drinking more than 5 days a week and 30% were drinking more than 4 units on the heaviest drinking day. Also, recent evidence reports an average AUDIT-C score in the general population of 2.75 [54], which is significantly lower than the score in our sample. 

#### 3.2.3. Type of Diet

DINE fiber intake was low (score < 30) in 117 (60%) individuals, medium (score between 30 and 40) in 43 (22%) individuals, and high (score > 40) in 34 (18%) individuals. DINE saturated fat intake was low (score < 30) in 121 (62%) individuals, medium (score between 30 and 40) in 51 (27%) individuals, and high (score > 40) in 22 (11%) individuals. DINE unsaturated fat intake was low (score < 6) in 19 (10%) individuals, medium (score between 6 and 9) in 107 (55%) individuals, and high (score > 9) in 68 (35%) individuals (Figure 2). 

We did not find comparable values for DINE scores in government statistics. 

#### 3.2.4. Physical Activity

Average days of vigorous physical activity were 1.21 ± 1.78 per week, during which the average activity was 43.06 ± 72.44 min per day. Average days of moderate physical activity were 1.73 ± 2.01 per week, during which the average activity was 43.06 ± 72.45 min per day. Average days of walking were 5.25 ± 2.16, during which the average activity was 94.47 ± 115.35 min per day. Average time spent sitting was 463.80 ± 278.40 min per week (Figure 3). 

Government statistics published in 2015 [55] showed that 66% of young adults (19–34 years) in England met the recommended physical activity guidelines (75 min of vigorous or 150 min of moderate activity, see discussion) for vigorous or moderate activity, compared to the significantly lower proportion of 47% (bootstrapped 95%CI 41–53%) in our sample.

Detailed physical health data of the sample are described in Table 2. 

#### 3.2.5. Physical Parameters

Averages in the sample were 71.53 ± 16.04 kg for weight (bootstrapped 95%CI 69.21–73.77), 1.72 ± 0.10 m for height (bootstrapped 95%CI 1.70–1.73), 24.45 ± 4.50 for BMI (bootstrapped 95%CI 23.94–25.09), 82.29 ± 13.04 cm for waist circumference (bootstrapped 95%CI 80.23–84.46), 69.42 ± 11.57 beats per minute for heart rate (bootstrapped 95%CI 67.72–71.13), 17.93 ± 5.32 acts per minute for respiratory rate (bootstrapped 95%CI 17.14–18.77), 115.81 ± 12.00 mmHg for systolic pressure (bootstrapped 95%CI 114.02–117.58), and 72.24 ± 9.32 mmHg for diastolic pressure (bootstrapped 95%CI 70.90–73.58).

This evidence is in line with average parameters collected in the general UK population (i.e., BMI = 24.6 [56], waist circumference = 82.5 cm [57]). Detailed physical parameters of the sample are reported in Table 3.

### 3.3. Gender Differences of Physical Health Data in CHR-P Samples

Tobacco use: differences in the number of smokers were nearly significant (*p* = 0.08) between male and female subjects (47% vs. 34% respectively).Alcohol use: differences were not significant for the number of drinkers (*p* = 0.60), AUDIT-C score (*p* = 0.63) and AUDIT total score (*p* = 0.63) between male and female subjects.Type of diet: differences between male and female subjects were non-significant for DINE total (*p* = 0.28), DINE fiber (*p* = 0.09), DINE fat (*p* = 0.10), DINE unsaturated fat (*p* = 0.69).Physical activity: difference between male and female subjects were significant (*p* < 0.001) for both days (1.80 ± 2.03 vs. 0.66 ± 1.37) and minutes-per-day of vigorous physical activity (70.20 ± 96.35 vs. 18.66 ± 39.34). There were no significant differences between male and female subjects regarding moderate activity, walking, and time spent sitting.Physical parameters: significative differences between male and female subjects were detected for weight (*p* < 0.001), height (*p* = 0.002), heart rate (*p* = 0.002), systolic pressure (*p* < 0.001), and diastolic pressure (*p* = 0.002). These differences reflect physiological differences that are also present in the general population [58].

## 4. Discussion

To our knowledge, the present study includes one of the largest CHR-P samples to date (*n* = 194) that has been investigated on physical health and lifestyle outcomes. We reported 41% of CHR-P individuals were tobacco smokers, almost double compared to the percentage in the UK general population of the same age. Overall, 49% of the subjects who consumed alcohol had an AUDIT-C score above 5 (hazardous drinking), with an average AUDIT-C score of 4.94, which is almost double of the average AUDIT-C score in the UK general population (2.75). Investigation of diet revealed low fiber intake in the majority of the sample and high saturated fat intake in 10% of the individuals. The results for physical activity showed a low proportion of subjects meeting the recommended physical activity guidelines (47% vs. 66% of young adults of the same age in UK).

Prevalence of tobacco smoking in individuals with schizophrenia is four to five times higher than the healthy population and over 60% of the patients are smokers [59], but smoking becomes a habit before the onset of schizophrenia in 77% of the cases [60] with an average anticipation of 11 years [61]. Hence, the onset of the smoking habit might coincide with the period in which the first symptoms of psychosis appear [62,63]. Our results of increased smoking habit in CHR-P individuals confirm this evidence and are in line with previous research from our team [2,64,65,66]. This CHR-P state thus provides an interesting framework to examine this association [67], constituting not only a phase in which smoking habit and attenuated psychotic symptoms coexist and influence each other, but also a window of opportunity to investigate specific reasons for initiating tobacco use and provide effective smoking cessation support [68]. In fact, on average, tobacco dependence was low-moderate (average FTND was 2.5), suggesting the CHR-P stage may be an optimal period for smoking cessation strategies to be effective, since they could be implemented before the development of a higher level of dependence, as shown in recent schizophrenic samples where FTND score was around 5 [69].

Regarding alcohol use in our sample, even if the proportion of individuals not consuming alcohol is comparable to the same age group in the UK general population, it is important to underline how both AUDIT-C and AUDIT total average scores in drinkers were almost coincident with the threshold score for hazardous alcohol use. If we also take into account the long-known issue of inconsistency related to self-reports of alcohol use in young adults [70], these averages may also be undervalued. This evidence confirms previous reports about higher alcohol consumption in CHR-P individuals and that an at-risk status is associated with alcohol involvement [71,72]. Of note, alcohol was also found as an important confounder between cannabis misuse and psychosis conversion in a high-risk sample [72]. These observations underline the importance of implementing specific alcohol abuse monitoring and prevention interventions in the CHR-P phase. 

Results from the investigation on the type of diet revealed that more than 60% of individuals in the sample had a low intake of fiber, while almost 10% of the sample had a high saturated fat intake. Since there is still scarce evidence in the literature about the use of the DINE questionnaire in the general population, our study is one of the first in the CHR-P literature to show standardized scores for specific types of diet intake. Also, our data show how diet could be targeted in these individuals to prevent other cardio-metabolic risk factors that are well known in psychosis and at-risk individuals [21,73]. 

UK national guidelines about physical activity published in 2021 [74] recommend at least 75 min of vigorous or 150 min of moderate activity every week; in our sample, averages were far below these values, with around 52 min of vigorous activity and 75 min of moderate activity. Of note, average vigorous physical activity was significantly different between male and female subjects (126 vs. 13 min), but also differences in moderate physical activity (174 vs. 70 min) were relevant, even if the difference was not statistically significant.

Among the limitations of our study, the lack of a control group was addressed by comparing our results with data from the UK census using bootstrapping. However, individuals participating in national surveys might not be help-seeking for mental health problems, like CHR-P individuals presenting to OASIS. Also, data from national surveys might not be collected in clinical examinations by doctors or other healthcare professionals. Finally, the cross-sectional design of our study does not explain causality (or lack of) between unhealthy lifestyles in CHR-P individuals and other important outcomes (i.e., risk of transition).

## 5. Conclusions

Our study, through the investigation of physical health and lifestyle outcomes in a large sample size of CHR-P individuals, showed that the percentage of smokers is 41% (twice as high as the general population) and those who consume alcohol have drinking behaviors that might be more dangerous and possibly lead to abuse and addiction. Diet was unbalanced, with high proportions of low fiber intake and high saturated fat intake, while subjects meeting the recommended physical activity guidelines were a low proportion (47% vs. 66% of young adults of the same age in UK). This evidence corroborates the need for monitoring physical health parameters and lifestyle in CHR-P subjects to increase our knowledge about their causes and implement tailored interventions for targeting daily habits. Even if high-quality research focused on physical health in young people with CHR-P is still scarce [29], interventions aimed at reducing alcohol and tobacco use, instead of promoting a balanced diet and physical activity that adheres to national guidelines, would constitute favorable and generalizable treatments in CHR-P, as they are effective towards comorbidities (i.e., depression) and not only for the individuals who will develop psychosis.

## Figures and Tables

**Figure 1 brainsci-13-00128-f001:**
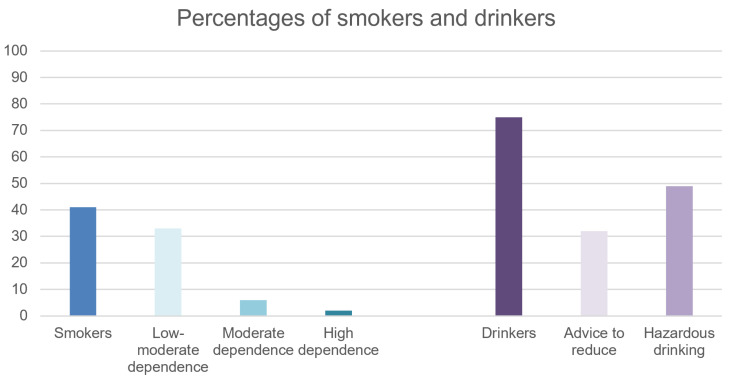
Percentages of smokers and drinkers.

**Figure 2 brainsci-13-00128-f002:**
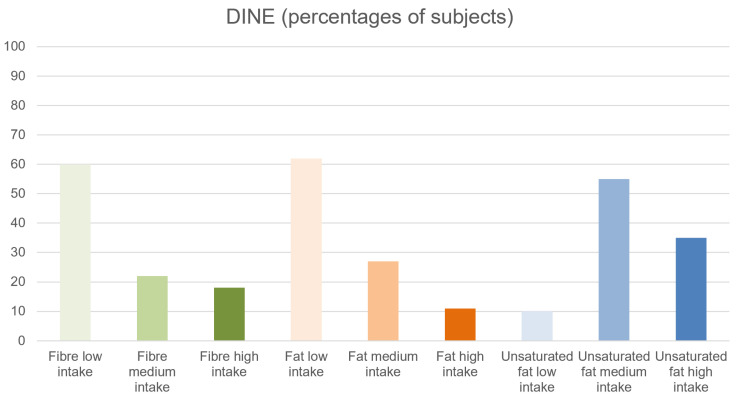
Results of DINE questionnaire (percentages of subjects).

**Figure 3 brainsci-13-00128-f003:**
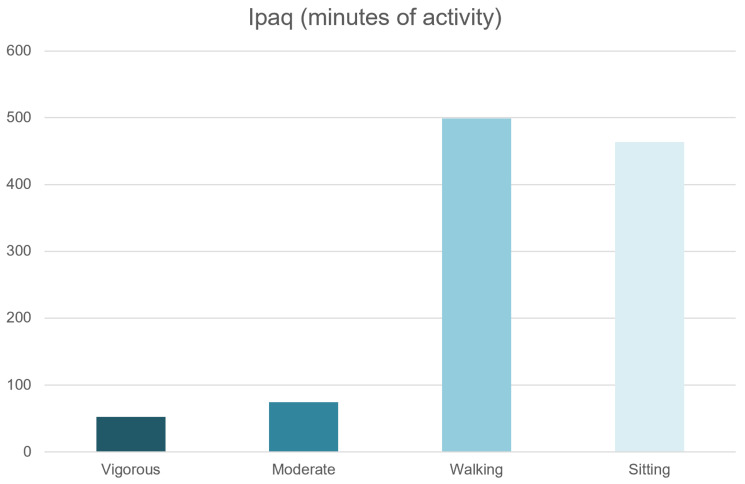
Results of IPAQ questionnaire (minutes of activity per week, calculated as “average days of activity per week” x “average minutes of activity per day”).

**Table 1 brainsci-13-00128-t001:** Sociodemographic parameters.

		N	Mean	Sd
Age (years)		194	23.70	5.12
		N	Count	%
Gender		194		
	Females		90	46
	Males		104	54
Ethnicity		174		
	White		72	41
	Asian		4	2
	Black African		16	9
	Black Caribbean		5	3
	Black British		37	21
	Other		40	23

**Table 2 brainsci-13-00128-t002:** Physical health data: tobacco use, alcohol use, type of diet, and physical activity.

	Total			Males			Females			Difference M/F
	N	Count	%	N	Count	%	N	Count	%	Fisher’s Exact Test
Tobacco smokers	194	80	41	104	49	47	90	31	34	*p* = 0.08
Alcohol drinkers	194	139	75	104	71	68	90	68	76	*p* = 0.58
	N	Mean	SD	N	Mean	SD	N	Mean	SD	Student’s *t* test
Daily cigarettes	77	8.47	7.86	48	9.69	8.97	29	6.98	5.65	t = 1.46; *p* = 0.15
FTND score	77	2.51	2.54	48	2.68	2.55	29	2.27	2.51	t = 0.70; *p* = 0.49
Audit-C score	139	4.94	2.93	71	5.39	3.08	68	4.47	2.70	t = 1.87; *p* = 0.63
Audit total score	139	7.88	6.63	71	8.90	6.99	68	6.81	6.10	t = 1.87; *p* = 0.63
DINE fibre score	187	28.05	14.02	99	29.60	15.28	88	26.07	12.32	t = 1.72; *p* = 0.09
DINE saturated fat score	187	27.22	11.57	99	28.57	11.25	88	25.83	11.83	t = 1.62; *p* = 0.10
DINE unsaturated fat score	187	8.68	2.39	99	8.61	2.27	88	8.75	2.54	t = −0.40; *p* = 0.69
IPAQ vigorous activity (dpw)	160	1.21	1.78	98	1.80	2.03	86	0.66	1.37	t = 4.38; *p* < 0.001 *
IPAQ vigorous activity (mpd)	160	43.06	72.44	98	70.20	96.35	86	18.66	39.34	t = 4.63; *p* < 0.001 *
IPAQ moderate activity (dpw)	160	1.73	2.01	99	2.11	2.12	85	1.55	1.99	t = 1.85; *p* = 0.66
IPAQ moderate activity (mpd)	160	43.06	72.45	97	82.37	144.77	84	45.42	83.65	t = 2.06; *p* = 0.41
IPAQ walking (dpw)	160	5.25	2.16	97	5.13	2.24	85	5.42	2.01	t = −0.91; *p* = 0.36
IPAQ walking (mpd)	160	94.47	115.35	94	94.41	127.13	83	107.23	123.11	t = −0.68; *p* = 0.50
IPAQ sitting (mpw)	160	463.80	278.40	91	446.40	282.60	75	484.20	260.40	t = −0.88; *p* = 0.38

dpw: days per week; mpd: minutes per day; mpw: minutes per week. Relative sample size was indicated for each variable only when missing data were present. *p* values are reported as unadjusted, statistical significance is reported as * and is set for this table at *p* = 0.004 (0.05/14) using the Bonferroni correction for multiple comparison.

**Table 3 brainsci-13-00128-t003:** Physical health data: physical parameters.

	Total			Males			Females			Difference M/F
	N	Mean	SD	N	Mean	SD	N	Mean	SD	Student’s *t* Test
Weight (Kg)	184	71.53	16.04	102	74.54	16.56	82	67.74	14.72	t = 2.91; *p* < 0.001 *
Height (m)	184	1.72	0.10	103	1.78	0.08	81	1.64	0.06	t = 12.75; *p* < 0.001 *
Body Mass Index	182	24.45	4.50	102	23.86	3.82	80	25.20	5.19	t = −2.02; *p* = 0.45
Waist circumference (cm)	144	82.29	13.04	83	83.00	11.73	61	81.33	14.68	t = 0.76; *p* = 0.45
Heart rate (beats per minute)	185	69.42	11.57	103	67.33	12.19	81	72.31	10.00	t = 2.97; *p* = 0.002 *
Respiratory rate (acts per minute)	165	17.93	5.32	93	17.22	4.95	71	18.96	5.65	t = 2.10; *p* = 0.37
Systolic pressure (mmHg)	178	115.81	12.00	98	120.20	10.76	79	110.34	11.33	t = 5.92; *p* < 0.001 *
Diastolic pressure (mmHg)	178	72.24	9.32	98	74.08	9.52	79	70.09	8.61	t = 2.89; *p* = 0.002 *

Relative sample size was indicated for each variable when missing data were present. *p* values are reported as unadjusted, significance is reported as * and is set at *p* = 0.006 (0.05/8) using the Bonferroni correction for multiple comparison.

## Data Availability

The authors give no permission to share raw data.

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
