# Peer review of "Physical Health in Clinical High Risk for Psychosis Individuals: A Cross-Sectional Study"

_brainsci, 2023, doi:10.3390/brainsci13010128_

Round 1

Reviewer 1 Report

This is a very interesting study focusing on physical health trajectories in clinical high risk for psychosis individuals. The paper is well-written and within the scope of the journal. Before considering it for publication, we recommend several minor changes.

Abstract

1-The main objectives of the study should be described in the abstract section. It seems that the main aim was to describe physical health trajectories to draw recommendations to improve it.

2-How were recorded data obtained? Please, describe it in the abstract section.

Introduction

1-The introduction section is brief. I recommend to expand it by describing, perhaps, separately, the social consequences, mental health consequences, and physical comorbidities in patients at risk for psychosis.

2-The main aim of the study should be described in a separate section: 1.1.

Material and methods

1- The first subsection of that section should be Participants and study design. I recommend to describe it in a joined section. 

2-I recommend to rename the section "Material" by "Assessment instruments". Variables derived from these assessments are presented later.

Results

1-I recommend to add a separate section about gender differences in physical activity, diet, etc. This is really important in terms of discussion of results. Differences in alcohol and tobacco use may explain the other differences? Are alcohol and other sustance use disorders potentially predicting diet and physical activity? PLease, discuss it in the discussion section.

Conclusions

1-Which kind of physical health interventions would be recommended? This should be explained in the conclusions section.

Author Response

This is a very interesting study focusing on physical health trajectories in clinical high risk for psychosis individuals. The paper is well-written and within the scope of the journal.

We thank the reviewer for the comments and positive feedback.

Before considering it for publication, we recommend several minor changes.

Abstract

1-The main objectives of the study should be described in the abstract section. It seems that the main aim was to describe physical health trajectories to draw recommendations to improve it.

2-How were recorded data obtained? Please, describe it in the abstract section.

While trying to maintain the strict 200-word limit for the abstract, we have included the required information (point 1 and 2) to satisfy the reviewer:

 “The primary outcome was to describe physical health assessed in CHR-P individuals and obtained accessing Electronic Health Records at the South London and Maudsley (SLaM) NHS Foundation Trust, UK (January 2013-October 2020).”

Introduction

1-The introduction section is brief. I recommend to expand it by describing, perhaps, separately, the social consequences, mental health consequences, and physical comorbidities in patients at risk for psychosis.

We thank the reviewer for the comments. We expanded, as suggested, the introduction section adding the mental health and social consequences in patients at high risk for psychosis in separated paragraphs, preceding the paragraph about physical comorbidities:

“The prevalence of CHR-P individuals in the community is still undefined [3] and, also, there is scarce knowledge about the outcomes of individuals who do not transition to psychosis [4], who might remain at a lower level of functioning compared to nonpsychiatric subjects [5]. In those individuals who will transition to psychosis, most will develop a schizophrenia spectrum disorder according to the DSM/ICD [6]. A 15-year follow-up study found that CHR-P individuals develop psychotic disorder up to 10 years after ini-tial presentation [7], in line with a risk of transition ranging from 65 to 79% at 10 years reported by other studies [8].55. In addition to attenuated psychotic symptoms (APS),  and brief limited intermittent psychotic symptoms (BLIPS) and genetic risk and deterio-ration syndrome (GRD) that define the construct [9], many CHR-P individuals often pre-sent other psychiatric comorbidities (i.e. anxiety, depression, substance abuse) which might be clinically debilitating [10, 11]. Suicidality might be increased [12].

A core feature of the CHR-P state is also the marked impairment in psychosocial functioning, associated with high-risk symptoms [13]. Social impairment in these indi-viduals tends to be resistant to pharmacological and psychosocial treatment [14], consti-tuting a predictor of longitudinal outcome [15, 16], which is also associated with a de-creased subjective quality of life [17, 18]. Further deficits have also been underlined in the domain of social cognition, within a context of widespread mild cognitive deficits, falling at an intermediate level between that of healthy individuals and those diagnosed as having schizophrenia [13].”

2-The main aim of the study should be described in a separate section: 1.1.

We created a new sub-section 1.1 named “Aims of the study”.

Material and methods

1- The first subsection of that section should be Participants and study design. I recommend to describe it in a joined section. 

2-I recommend to rename the section "Material" by "Assessment instruments". Variables derived from these assessments are presented later.

We created a first (joined) sub-section named “Participants and study design” and renamed the following sub-section in “Assessment instruments”.

Results

1-I recommend to add a separate section about gender differences in physical activity, diet, etc. This is really important in terms of discussion of results. Differences in alcohol and tobacco use may explain the other differences? Are alcohol and other sustance use disorders potentially predicting diet and physical activity? PLease, discuss it in the discussion section.

We thank the reviewer for the comments. We added a separate section (3.3) about gender differences in physical health data, as suggested, in the results. Unfortunately, we did not find significative gender differences which were not expected: significative differences were related to the amount of vigorous physical activity, weight, height, heart rate and systolic/diastolic pressure. These differences reflect discrepancies which are also present in the general population [1][2]

We also conducted additional correlation analyses to investigate the potential influences suggested by the reviewer, and appended them in Supplementary Material (eTable 3). As above, relevant correlations were all expected (i.e., number of cigarettes smoked with FTND score; AUDIT-C with AUDIT total score; BMI with waist circumference) and therefore not useful to further explore our outcomes, but they confirmed the thorough collection of the parameters.

Conclusions

1-Which kind of physical health interventions would be recommended? This should be explained in the conclusions section.

Thank you, we added the physical interventions recommended in the conclusions section:

“Even if high-quality research focused on physical health in young people with CHR-P is still scarce [29], interventions aimed at reducing alcohol and tobacco use, instead promoting a balanced diet and physical activity that adheres to national guidelines, would constitute favourable and generalizable treatments in CHR-P, as they are effective towards comorbidities (i.e., depression) and not only for the individuals who will develop psychosis.”

References

  1. Prabhavathi, K., et al., Role of biological sex in normal cardiac function and in its disease outcome - a review. J Clin Diagn Res, 2014. 8(8): p. Be01-4.
  2. McCarthy, C., Warne, J.P. Gender differences in physical activity status and knowledge of Irish University staff and students. Sport Sci Health18, 1283–1291 (2022). https://doi.org/10.1007/s11332-022-00898-0

Reviewer 2 Report

This an interesting study dealing with helathy life-styles in subjects high risk for psychosis.

However more statistical analyses are required to improve the relevance of this study:

1. Is there any significant relationship between IPAQ questionnaire data and DINE questionnaire data?

2. Please analyse any correlation between IPAQ values and:

Body Mass Index

Waist circumference

Heart rate

Respiratory rate

Systolic pressure

Diastolic pressure

3. Please analyse any correlation between DINE values and:

Body Mass Index

Waist circumference

Heart rate

Respiratory rate

Systolic pressure

Diastolic pressure

Author Response

This an interesting study dealing with helathy life-styles in subjects high risk for psychosis.

However more statistical analyses are required to improve the relevance of this study:

  1. Is there any significant relationship between IPAQ questionnaire data and DINE questionnaire data?
  2. Please analyse any correlation between IPAQ values and:

Body Mass Index

Waist circumference

Heart rate

Respiratory rate

Systolic pressure

Diastolic pressure

  1. Please analyse any correlation between DINE values and:

Body Mass Index

Waist circumference

Heart rate

Respiratory rate

Systolic pressure

Diastolic pressure

We thank the reviewer for the comments. We conducted all the statistical analyses suggested and we included them in the supplementary material. Unfortunately, relevant correlations (Pearson’s r >|0.3|) were all expected and therefore not useful to further explore our outcomes, but they confirmed the thorough collection of the parameters.

1. eTable2a.  Correlation matrix between DINE scores and IPAQ data.
Relevant correlations (Pearson’s r >|0.3|)  were expected and involved only the different sub-scales of the IPAQ questionnaire (i.e., individuals who spent more time sitting, spent less time doing vigorous or moderate activity).

  1. eTable2b. Correlation matrix between IPAQ data and physical parameters
    Relevant correlations (Pearson’s r >|0.3|) were all expected and involved:
  • the different sub-scales of the IPAQ questionnaire (i.e., individuals who spent more time sitting, spent less time doing vigorous or moderate activity).
  • Waist circumference is positively correlated with BMI and systolic pressure; Systolic pressure is positively correlated with diastolic pressure

  1. eTable2c. Correlation matrix between DINE scores and physical parameters

Relevant correlations (Pearson’s r >|0.3|) were all expected: Waist circumference is positively correlated with BMI and systolic pressure; Systolic pressure is positively correlated with diastolic pressure

Round 2

Reviewer 2 Report

Agreed to revised version.